# Understanding public opinion to the introduction of minimum unit pricing in Scotland: a qualitative study using Twitter

Laurence Astill Wright, [ORCID] Su Golder, Adam Balkham, J McCambridge

## ABSTRACT

**Objectives** On 1 May 2018 minimum unit pricing (MUP) of alcohol was introduced in Scotland. This study used Twitter posts to quantify sentiment expressed online during the introduction of MUP, conducted a thematic analysis of these perceptions and analysed which Twitter users were associated with which particular sentiments.

**Design and setting** This qualitative social media analysis captured all tweets relating to MUP during the 2 weeks after the introduction of the policy. These tweets were assessed using a mixture of human and machine coding for relevance, sentiment and source. A thematic analysis was conducted.

**Participants** 74 639 tweets were collected over 14 days. Of these 53 574 were relevant to MUP.

**Results** Study findings demonstrate that opinion on the introduction of MUP in Scotland was somewhat divided, as far as is discernible on Twitter, with a slightly higher proportion of positive posts (35%) than negative posts (28%), with positive sentiment stronger in Scotland itself. Furthermore, 55% of positive tweets/retweets were originally made by health or alcohol policy-related individuals or organisations. Thematic analysis of tweets showed some evidence of misunderstanding around policy issues.

**Conclusions** It is possible to appreciate the divided nature of public opinion on the introduction of MUP in Scotland using Twitter, the nature of the sentiment around it and the key actors involved. It will be possible to later study how this changes when the policy becomes more established.

## BACKGROUND

Over the last 40 years the relative price of alcohol has decreased significantly in many countries throughout the world. Alcohol has never been as widely available and affordable as it currently is and this is primarily due to taxation falling behind increased earnings and inflation.[1] To combat the 3.3 million deaths worldwide each year and 5.1% of the global burden of disease, the WHO recommends appropriate taxation and pricing policies in order to increase the cost of alcohol

Department of Health Sciences, University of York, York, UK

**Correspondence to**
Dr Laurence Astill Wright; laurencewright@doctors.org.uk

## Strengths and limitations of this study

► This is the largest social media study conducted on alcohol policy with analysis of 53 574 relevant tweets.
► This is the first alcohol policy study to use a mixture of human and machine classification.
► Using the Twitter firehose and 29 synonyms in our search string maximised the number of Tweets collected.
► Classification was not perfect but agreement between coders was very good.
► Twitter is not representative of the general population.

as part of an overall public health strategy to reduce harmful drinking.[2 3]

Increasing alcohol prices consistently reduces consumption[4] and a minimum unit price (MUP) of 50p in Scotland is forecast to decrease the consumption of harmful drinkers by 7%, hazardous drinkers by 2.5% and moderate drinkers by 1.2%.[5] Changes in taxation alone would require a 70% increase to cause a reduction of 7% in consumption by harmful drinkers.[5] In comparison, MUP specifically targets the cheapest drinks favoured by the heaviest drinkers.[4]

In the second half of the 20th century Scotland has struggled with the increasing health, social and economical consequences of greater alcohol consumption more so than the rest of the UK.[6] Average weekly unit consumption and rates of chronic liver disease and cirrhosis are higher than in England and Wales.[6] There are significant economical costs in healthcare provision, crime and lost productivity.[7] It is predicted that MUP will reduce the number of deaths due to alcohol by 60, hospital admissions by 1300 and crimes by 3500 in the first year alone.[5]

After a series of legal challenges and national debate lasting approximately 5 years,

MUP was approved by the UK Supreme Court and was introduced to Scotland on 1 May 2018.[8] It has been demonstrated that alcohol industry submissions made to a Scottish government consultation in 2008 misrepresented the peer-reviewed literature surrounding alcohol policy.[9] The arguments made against MUP during this consultation, such as the concern of a new black market alcohol industry, were reiterated by Scottish and UK newspapers in 2011 and 2012. Some newspapers argued that MUP would be ineffective and it would punish responsible drinkers and the poor, while those that advocated for MUP argued that it would reduce health and social harms.[10]

Public perception appears to be changing over time on MUP with the 2015 British Social Attitudes survey suggesting that 52% of British adults support MUP, 25% are against it and 22% are unsure,[11] compared with a British 2011 YouGov survey which suggested 47% and 44% for and against it, respectively, while 9% were unsure.[12] A 2011 focus group study had suggested that British participants held largely negative attitudes towards MUP due to 'a misunderstanding of the minimum price per unit policy itself' and 'the failure to recognise the significance of small incremental reductions in alcohol consumption'.[13] A further focus group study identified beliefs consistently associated with negative attitudes of pricing policy[14]; that pricing policies will make no difference to behaviour, the government considers the national economy to be more important than the health of the general public and that government cannot be trusted.[14]

When comparing the popularity of alcohol policy approaches in a discrete choice experiment, Pechey *et al*[15] demonstrate that MUP is less favoured than both regulating alcohol marketing and decreasing the number of alcohol sales outlets when consumption, health and social outcomes are not considered. However, Pechey *et al*[15] do show that the popularity of MUP increases from 43% to 63% when considering its significant effects in reducing consumption and social harms. In the UK there are no robust relationships between socio-economic status and support for alcohol policy options.[16] In other countries it has been found that heavier drinkers (whose drinking is most damaging to themselves and others) are less supportive of alcohol policy change.[17 18] Pechey *et al*[15] suggest that policymakers should focus on the beneficial outcomes when advocating for MUP to increase public support.

Few studies have used online social media to try to ascertain public attitudes towards change in alcohol policy. Twitter is a social networking website where users can broadcast their opinions to a public audience. As of late 2017, Twitter had 330 million active monthly users[19] and has great potential as a resource for quantitative and qualitative analysis of public opinion. Stautz *et al*[20] analysed the reaction to the updated UK alcohol guidelines in 2017, identifying that the majority of tweets were unsupportive of the adjustments, which reduced the advised limits for low risk drinking downwards for men, and that the community as a whole was largely opposed to alcohol policy measures. No other studies have attempted to assess public reaction to alcohol-related policy changes using Twitter, although studies have assessed the perception of cannabis use,[21] electronic cigarettes[22] and electronic cigarette marketing.[23]

This study used Twitter posts to quantify sentiment expressed online during the introduction of MUP, conducts a thematic analysis of these perceptions and analyses which Twitter users are associated with which particular sentiments. Our specific research questions were as follows: (1) What are the proportions of positive, negative and neutral tweets? (2) What themes are commonly expressed? (3) Which Twitter users are expressing which themes? (4) Do the results mirror population survey data and other qualitative research surrounding MUP?

## METHODS

Data were collected from Twitter using the Gnip Power-Track firehose provided by DiscoverText (a text analytics software - https://discovertext.com). DiscoverText, which has been used in previous Twitter research,[23] was also used to archive and machine-code tweets. Data collection started on 29 April 2018, with the introduction of MUP on 1 May 2018. Due to the large volume of tweets collected, data collection was stopped at 14 days ending on 12 May 2018. Research methods were in accordance with Rivers' and Lewis'[24] recommendations for the ethical use of Twitter data.

Search terms were trialled using Twitter's free search application programming interface (API - https://twitter.com/search-advanced). Terms that produced more than one search result relating to MUP on the first page of search results were included. Different terms and spellings were trialled, and hashtags that were repeatedly mentioned in tweets were included. Slang terms for alcohol were identified using online thesauruses (eg, www.urbandictionary.com) and promising search terms included. Only English language tweets were included. The Twitter firehose was used to collect all publicly available tweets corresponding to the relevant search terms without the limitations of the Twitter API.

The final search strategy was:

((minimum unit price) OR (minimum unit pricing) OR (minimum pricing) OR (minimum price) OR (minimum alcohol price) OR (minimum alcohol pricing) OR (minimum booze price) OR (minimum booze pricing) OR (min booze price) OR (min booze pricing) OR (min unit price) OR (min unit pricing) OR (min alcohol price) OR (min alcohol pricing) OR (MUP) OR (50p unit) OR (Scotland alcohol) OR (Scotland booze) OR (Scotland bevvy) OR (Scotland min price) OR (Scotland min pricing) OR (alcohol unit) OR (minimum price per unit) OR (cheap booze Scotland) OR (minimum cost alcohol) OR (min cost alcohol) OR #minimumunitpricing OR #mupsaveslives OR #MUP) lang:en

Tweets were initially coded as relevant or irrelevant by a single human coder. The human coder endeavours were used to train DiscoverText's machine classifier using a Naive Bayes algorithm. This allows machine coding of remaining tweets and was then applied to the full selection of tweets. While the algorithm was excellent at excluding irrelevant tweets, sometimes these irrelevant tweets were incorrectly classified as relevant. This was an iterative process and so the machine classifier was retrained until it reached an acceptable degree of accuracy. The agreement between machine classifier and human coder (LAW) was calculated using a kappa score on an overlapping selection of 100 tweets. The human coder's work was validated against a second human coder (AB) on an overlapping selection of 500 tweets. Irrelevant tweets were then discarded.

Once relevant tweets were separated from irrelevant, a similar process was used to classify tweets according to sentiment. A single human coder classified relevant tweets into positive, negative and neutral. The coding of the primary coder was validated against the coding of the second human coder using a kappa score on an overlapping sample of 200 tweets. Series of 200 tweets were double classified until kappa scores greater than 0.7 were achieved and this was used to train a new custom machine classifier that was applied to all the relevant tweets. Inaccuracies in machine coding were refined by human coding of key tweets to retrain the algorithm. Again, machine coding was validated using a kappa score on an overlapping sample of 100 tweets.

Once relevant tweets were separated into positive, negative and neutral, a random sample of 500 tweets was taken from each of the three subgroups using DiscoverText's random sampling tool. These 1500 tweets were analysed and single coded to assess the predominant themes. Prior to assessment we reviewed previous media arguments for and against MUP,[10] and various public surveys[25] to establish the range of anticipated themes (here we identified four positive themes and eight negative themes). The subsequent process of single coding to assess the predominant themes was an iterative process and when a theme was not congruent with the anticipated themes, it was considered a newly emerging theme and this was added. Two new themes emerged through this process, one positive and one negative.

New themes, in addition to those already identified, emerged only in the initial stages of analysis and no new themes emerged in the later stages of analysis (the final 150 tweets of each 500 tweet sample). Thus it was determined that sufficient saturation had been reached, and no additional tweets needed to be examined. The popularity of each theme was also assessed in each random sample.

These three random samples of 500 tweets were also analysed to determine the source of the tweets. A single human coder examined each author's Twitter page. For each tweet/retweet, the username, full name, associated biography and the associated results from an internet search engine (https://www.google.com) were examined to determine the user's background. The same process was used to determine if the source self identified as Scottish or lived in Scotland or not (only 1.6% of Twitter users have their geolocation activated and so inferences must be made from their profile).[26] For example, Twitter profiles contain a space for a user to write their location and if this was a Scottish place it was assumed that the user was Scottish. Some users did not write a location but had explicit references to the place they lived in their tweets. $X^2$ tests were used to determine if any differences in proportions reached statistical significance (p value<0.05) in categorical variables.

## PATIENT AND PUBLIC INVOLVEMENT

There was no patient and public involvement in the design or conduct of this study. As participants did not explicitly consent to their Tweets being used in this specific research paper, paraphrased examples of Tweets were used to retain anonymity.

## RESULTS

In total, 74 639 tweets were collected over 14 days, 62 879 of these tweets were manually coded as either 'relevant' to MUP or 'not relevant' by the same coder (LAW) and the Naive Bayes algorithm subsequently coded the remaining 11 760 tweets. Five hundred tweets were coded by both the primary coder and a second coder (AB). Of these 500 tweets, there was a 97% agreement with a kappa score of 0.95. This indicates an excellent level of agreement. In order to validate the coding of the algorithm, 100 tweets were coded by both the primary coder (LAW) and the algorithm. For these 100 tweets there was a 97% agreement with a kappa score of 0.94, providing further reassurance about reliability on relevance.

Of the 74 639 tweets, 53 574 (72%) tweets were classified as relevant, while 21 065 (28%) were classified as 'not relevant'. The irrelevant tweets made no reference to the MUP of alcohol in any context. These 53 574 relevant tweets were subsequently classified according to sentiment and 57 801 tweets were manually coded among which 18 741 were coded as positive (35%), 14 866 as negative (28%), 17 302 as neutral (32%) and 2665 as not relevant (5%). In the 200 tweets coded by both the primary and secondary coder there was a kappa score of 0.75. The kappa scores were: positive - 0.79, negative - 0.74, neutral - 0.76, not relevant - 0.73. This shows good agreement for sentiment tweets. For 100 tweets coded by both the primary coder (LAW) and the algorithm there was a 96% agreement with a kappa score of 0.94.

From each sentiment (positive, negative and neutral) 500 randomly selected tweets were analysed for predominant themes. These were elaborated through the process of thematically coding each tweet and new themes were added as they occurred until saturation was reached. Twitter based thematic analysis is difficult to automate

**Table 1** Results of thematic analysis of positive, negative and neutral tweets with paraphrased examples

| | n | % | Paraphrased example |
|---|---|---|---|
| **Theme of positive tweets** | | | |
| Reduces health harms | 352 | 70.4 | Minimum unit pricing will decrease hospital admissions and save lives #mupsaveslives |
| Reduces social harms | 13 | 2.6 | This will greatly reduce alcohol-fuelled violence and other countries must follow |
| Effectively targets the cheapest, strongest alcohol | 36 | 7.2 | Strong cider sold at pocket money prices is hugely damaging |
| Scotland has an alcohol problem and something must be done | 26 | 5.2 | This country has an awful relationship with drink - let's try MUP |
| MUP is an evidence-based policy | 5 | 1.0 | The evidence backs MUP, which has been approved by the courts and will be extensively evaluated with a sunset clause |
| Nil reason given | 60 | 12.0 | Excellent work from the SNP! |
| Incorrectly classified as positive | 8 | 1.6 | |
| Total | 500 | 100 | |
| **Theme of negative tweets** | | | |
| Alcoholics will not decrease their alcohol intake | 138 | 27.6 | Alcoholics will not buy less but their children will go without so they can get it |
| Increase in illicit alcohol production and/or encourage cross-border trading | 71 | 14.2 | Hoards will rush over the border to stock up on frosty jacks - who would've thought we'd have a booze cruise in 2018 |
| Libertarian | 54 | 10.8 | First the sugar tax and now this - the nanny state won't stop |
| A tax on the poor | 52 | 10.4 | Another example of a classist poor-bashing policy |
| Increase in drug use and/or petty crime | 23 | 4.6 | Neds will rob grannies for booze money and the jakeys will turn to drugs instead |
| Punishes responsible drinkers | 17 | 3.4 | A few people can't drink responsibly and now everyone else has to pay the price? |
| Increases retailer profits | 6 | 1.2 | All this will do is line the pockets of billionaires - the supermarkets can't believe their luck |
| Harms businesses | 2 | 0.4 | How many jobs will be lost from this? |
| Alcohol consumption is a cultural problem | 3 | 0.6 | Other countries with cheap alcohol don't have the same problems - the problem isn't to do with the price |
| Nil reason given | 108 | 21.6 | This new alcohol law is embarrassing bs #SNPfail |
| Incorrectly classified as negative | 26 | 5.2 | |
| Total | 500 | 100 | |
| **Theme of neutral tweets** | | | |
| Factual | 301 | 60 | Scotland introduces new alcohol law |
| Humour | 102 | 20 | Great that Scotland are adopting the alcohol pricing design they have trialled for so long at the Edinburgh fringe |
| Balanced/Unclear Sentiment | 82 | 16 | On the one hand it could reduce overconsumption of alcohol, but on the other it could encourage a black market |
| Incorrectly classified as neutral | 15 | 3 | |

MUP, minimum unit pricing; SNP, Scottish National Party.

using machine algorithms due to abbreviations, emoticons and sarcasm[27] and we relied exclusively on human coding. The findings are presented in table 1. Perceived ability to reduce health harms was the most prominent theme in the positive tweets, scepticism about effects on problem drinkers was the most prominent in the negative tweets and factual information were the most prominent theme in the neutral tweets. In each of these sentiment categories a small proportion of tweets (ranging from 1.6% to 5.2%) were found to be misclassified. Some were irrelevant but falsely classified as relevant, while some were of another sentiment. It is impossible to say whether it was human or machine coding which produced this error.

The random samples of 500 tweets divided by sentiment were next classified according to the background of the Twitter user who posted the tweet. In the case of retweets this was the original tweeter. It was not possible to determine who retweets were made by. The users were divided up into the groups as in table 2. Miscellaneous users were those accounts who did not fall into any of the other groups and largely consisted of private companies and spam accounts.

**Table 2** Source analysis of tweets and retweets from positive, negative and neutral subgroups

| | | Original source of tweet or retweet | | | | | | |
| | | Member of public | Health/alcohol policy organisation/individual | Media/news organisation/individual | Alcohol industry-related organisation/individual | Celebrity/public figure | Miscellaneous | Incorrectly classified as positive/negative/neutral | Total |
|---|---|---|---|---|---|---|---|---|---|
| Positive | n | 85 | 275 | 91 | 1 | 17 | 25 | 6 | 500 |
| | % | 17 | 55 | 18.2 | 0.2 | 3.4 | 5.0 | 1.2 | 100 |
| Negative | n | 287 | 53 | 56 | 5 | 2 | 81 | 16 | 500 |
| | % | 57.4 | 10.6 | 11.2 | 1.0 | 0.4 | 16.2 | 3.2 | 100 |
| Neutral | n | 189 | 45 | 109 | 18 | 10 | 120 | 9 | 500 |
| | % | 37.8 | 9.0 | 21.8 | 3.6 | 2.0 | 24 | 1.8 | 100 |

Table 3 presents data on source of the tweets, with more positive sentiment demonstrated among those which were likely Scottish. To further examine these data, we used a $X^2$ test and found that there was a significant difference in sentiment between likely Scottish and not obviously Scottish Twitter accounts ($X^2$ statistic 22.659, df=2, p value<0.001).

## DISCUSSION

Study findings demonstrate that public opinion on the introduction of MUP in Scotland was somewhat divided, with a slightly higher proportion of positive posts (35%) than negative (28%) or neutral (32%). This was the case particularly in Scotland. These findings mirror previous survey data that suggest a growing proportion of the British public favour MUP than are against it.[11]

Public opinion alone does not dictate alcohol policy and there is often significant industry and political will to resist change. There do, however, remain complex interactions between public opinion and shifts in alcohol policy. Österberg and colleagues[28] demonstrated that a decrease in alcohol excise duty in Finland in 2004 and a subsequent rise in alcohol related harm led to an increase in support for alcohol policies to counteract these trends. In Ireland high levels of alcohol consumption and a doubling of alcohol related street violence over 7 years led to public discussions which culminated in increased alcohol taxation, via increased support for alcohol policies.[29] There is some suggestion in this study and the literature on which it draws that Scotland has followed a similar pattern to Finland and Ireland where it appears that an increase in alcohol harms has prompted public discussion putting alcohol policy change on policy agendas. A more nuanced historical study would be needed to investigate how far this is true, and the roles of political actors in relation to public opinion.[30 31]

In 1984 John Kingdon proposed that shifts in public policy require the overlapping of three different factors - the public acknowledgement of a problem, a clear solution to a problem and also the political will to address the issue.[32] In relation to MUP there was first a public discussion of the harms of alcohol consumption. Researchers and public health experts subsequently paid more attention to restrictive alcohol policies as a solution to this problem, and then the Scottish National Party showed the political will to address these alcohol harms.[30 31] These three factors may have overlapped to create a unique 'window of opportunity' to introduce MUP.

Only one other study has examined social media responses (analysing 3061 tweets) to alcohol policy-related developments.[20] The present study is thus the largest conducted on alcohol policy with analysis of 53 574 relevant tweets and the first to use a mixture of human and machine classification. Stautz et al[20] showed a predominantly negative reaction to updated alcohol guidelines (27.4% negative vs 6.8% positive).

**Table 3** Source nationality analysis of tweets and retweets from positive, negative and neutral subgroups

| | | Source nationality | | | | |
|---|---|---|---|---|---|---|
| | | Likely Scottish | Not obviously Scottish | Unable to view profile | Not correctly assigned the right sentiment | Total |
| Positive | N | 293 | 187 | 14 | 6 | 500 |
| | % | 58.6 | 37.4 | 2.8 | 1.2 | 100 |
| Negative | N | 218 | 204 | 65 | 13 | 500 |
| | % | 43.6 | 40.8 | 13 | 2.6 | 100 |
| Neutral | N | 219 | 259 | 13 | 9 | 500 |
| | % | 43.8 | 51.8 | 2.6 | 1.8 | 100 |

There are several interpretations of the large difference in sentiment between the updated alcohol guidelines and the introduction of MUP. While it is possible that public support for MUP is far greater, it is also possible that because Stautz et al[20] only followed one hashtag (while we followed 29 different synonyms accounting for different terminology and spellings), this procedure yielded a less representative sample. In our study the most popular hashtag relating to MUP was only used in 3.8% of relevant tweets and so we would not recommend searching based on hashtags alone when conducting future alcohol policy-related research.

Thematic analysis of positive tweets showed less variation in arguments supporting MUP than against it. Positive tweets focused on the health benefits of MUP was 70.4% and a minority focused on other views. This reflects the introduction of MUP for primarily public health reasons. Furthermore, health/alcohol policy organisations/individuals tweets or retweets were the original sources of the majority of positive tweets surrounding MUP. This suggested a coordinated response by public health organisations focusing on a single message - that MUP reduces alcohol-related health problems. These findings suggest implications for advocacy groups investing in social media to influence public opinion.

Additionally, our study found that 24.9% of the 1500 randomly selected sentiment tweets were made by health/alcohol policy-related Twitter accounts, while Stautz et al[20] demonstrated 12.4% of tweets relating to the updated UK alcohol guidelines were made by health related individuals/organisations. The responses of health advocacy groups to MUP appears to be more effective than that to alcohol guidelines, in part because of the ongoing failure to implement UK public information campaigns on the new guidelines, and the remarkable refusal of alcohol producers to carry the revised guidelines on alcohol packaging. As Pechey et al[15] recommend, Scottish policy-makers have given prominence to the expected outcomes of MUP.

Many of the negative themes expressed were similar to alcohol industry framings of the issues from earlier on in the public debate.[9 33] Following on from the industry's attempts to obstruct the implementation of MUP through legal processes, the alcohol actors we identified through Twitter continued to propagate the negative framing of MUP in an attempt to marginalise those arguments based on peer-reviewed literature. Yet, by the time of implementation of the policy, it is striking how little such activity there was by industry actors. It seems more likely that alcohol industry actors pursued other avenues to alter public perception post-MUP implementation rather than that they were inactive, and these were not captured in this study.

The similarity we have demonstrated in findings between Twitter-based research to gauge public perceptions and general population surveys may provide some support for social media-based methods as adjuncts to survey based opinion polling. As set out by our research questions, we showed that 35% of tweets were positive, 28% were negative and 32% were neutral. Similarly the 2011 YouGov survey suggested that 47% supported MUP, while 44% opposed it and 9% were unsure.[12] Likewise the 2015 British Social Attitudes survey suggests that 52% of British adults support MUP, while 25% are against it and 22% are unsure.[11] Gauging public opinion via social media has numerous practical advantages over polling, though validation methods remain to be developed. There will probably be lower costs given that the data are already in the public domain, and machine algorithms can be used to code items with high inter-rater reliability with human coders. Social media research does, however, bring with it new ethical challenges that must be considered by future researchers.[34]

Using the DiscoverText software, we were unable to distinguish between original tweets and retweets. It is likely that a significant proportion of the tweets were retweets, but we are unable to gauge what proportion, and this remains a limitation of our study. While retweets are perceived by many as an expression of agreement with the original tweet, this is not always the case. On occasion, retweets were accompanied by a comment from the user. In these circumstances the sentiment of the extra comment was analysed primarily, rather than the sentiment of the retweet.

Other limitations of our work include uncertainty about the inferences about public opinion that can be made

from this data. Twitter users are unlikely to be representative of the general population, as they are more likely to be urban dwelling, male and have higher educational achievement.[35 36] Twitter users tend to hold more extreme views[37] and surround themselves with those who hold similar opinions (in what is known as echo-chambers).[38] This makes any inferences in relation to previous polling data questionable. Furthermore, classification was not perfect and 3.3% of tweets were included in the wrong sentiment group in our random sample of 1500 tweets. It is also possible our results were subject to confounding, for example, by political affiliation. Many accounts provided limited biographical information and so this was not measured or adjusted for. In addition, while we demonstrated a high proportion of positive posts, this may not necessarily translate into behaviour change, or speak directly to the possible success of the policy. Nonetheless, it is possible to appreciate the divided nature of public opinion on the introduction of MUP, the nature of the sentiment around it and key actors involved, and it will be possible to later study how this picture changes when the policy becomes more established.

**Contributors**  LAW, SG and JM were responsible for the original study design. LAW was responsible for primary data coding, analysis and for initial drafting of this report. AB was the second coder. LAW, SG, AB and JM were responsible for subsequent interpretation, editing and rewriting for the report.

**Funding**  This work was supported by a Wellcome Trust Investigator Award in Humanities and Social Science (200321/Z/15/Z) to JM.

**Competing interests**  None declared.

**Patient consent for publication**  Not required.

**Ethics approval**  The University of York Health Sciences Research Governance Committee was consulted and recommended that the study did not require ethical approval as the Twitter data used were already in the public domain.

**Provenance and peer review**  Not commissioned; externally peer reviewed.

**Data sharing statement**  No additional data for this article is available.

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
