## [Reviewer comments · BMJ Open]

ARTICLE DETAILS

TITLE (PROVISIONAL)	Understanding public opinion to the introduction of Minimum Unit Pricing in Scotland: a qualitative study using Twitter
AUTHORS	Astill Wright, Laurence; Golder, Su; Balkham, Adam; McCambridge, J

VERSION 1 - REVIEW

REVIEWER	Kim Holmberg University of Turku, Finland
REVIEW RETURNED	11-Mar-2019

GENERAL COMMENTS	The authors present an interesting manuscript mining the public opinion on Twitter. The paper is very well written and structured, the methods and the results are presented in detail, and the results and possible weaknesses are adequately discussed. I assume that by using the firehose to retrieve the tweets the software used was able to collect all tweets matching the search terms, without any limitations that apply when using the free Twitter API? This could be mentioned in the manuscript. It was a bit difficult to follow how the coding, either by humans or machine, of the (many) different samples was done. Perhaps a graph or a flowchart would make the process clear? The manuscript doesn't distinguish between original tweets and retweets, but I'd argue that there is a difference between them. A retweet, for instance, doesn't necessary mean that the tweeter agrees with the original tweet. What was the proportion of retweets? It would be interesting to see if the results would change significantly if only original tweets were coded, but at least I think the authors should raise this question in the discussion, perhaps as a possible limitation of the study. While I think that the results of this study do answer the research questions, it would benefit the reader if the results were better linked to the respective research questions, i.e. the authors could and probably should return to the research questions in the end.
--

REVIEWER	Anurag Sharma University of New South Wales, Sydney, Australia
REVIEW RETURNED	14-Mar-2019

GENERAL COMMENTS	Sin taxes are gaining popularity among policy makers as a tool for behavioural change to achieve better health outcomes in the
--

	population. However, the response to such policies can be varied due to various stakeholders involved. In particular contrasting claims are made regarding general public support for such taxes. Thus there is a need for more evidence to gauge public perception of such policies. This paper helps fill this gap by using data from social media (twitter). Data from tweets related to the MUP policy were collected over two weeks after the policy was implemented in Scotland. Despite some obvious issues related to the representativeness of such data (adequately discussed in the limitations section) 35% of tweets were found to be positive compared to 28% negative tweets. My comments:  1. Pairwise group mean t-tests could be performed to see if the difference between positive, negative and neutral tweets is statistically significant. Similarly, groupings can be done for nationality and CIs reported in the table. 2. The fact that only 28% comments could be perceived as negative is the main takeaway from the results and should be emphasised in the abstract and discussion section. 3. Page 5 line 8-12 : Discussion not clear especially: 4 positive, 8 negative 4. More discussion/analysis needs to be done if negative tweets were backed by industry lobby groups explicitly/implicitly (through proxies) and if not what could be the possible reasons behind it. 5. The inference that industry groups were relatively inactive post implementation of policy could not be made just by twitter data analysis as they might be pursuing other avenues to change public perception. 6. It would have been interesting to see if there was surge in suggestions on the social media to get around MUP and somehow consume cheaper substitutes through border crossing etc. That implies positive reception of policy may not necessarily led to behaviour change. This issue is critical for the relevance of the findings: even if it is established this policy was well supported, will this lead to change in behaviour and reduced alcohol consumption. The last section should include discussion around this theme.
--	---

VERSION 1 – AUTHOR RESPONSE

Reviewer(s)' Comments to Author:

Reviewer: 1

Reviewer Name: Kim Holmberg

Institution and Country: University of Turku, Finland

Please state any competing interests or state 'None declared': None declared

Please leave your comments for the authors below

The authors present an interesting manuscript mining the public opinion on Twitter. The paper is very well written and structured, the methods and the results are presented in detail, and the results and possible weaknesses are adequately discussed.

I assume that by using the firehose to retrieve the tweets the software used was able to collect all tweets matching the search terms, without any limitations that apply when using the free Twitter API? This could be mentioned in the manuscript.

We did use the twitter firehose to capture all of the Tweets matching the search terms, We have now clarified this in the methods section.

It was a bit difficult to follow how the coding, either by humans or machine, of the (many) different samples was done. Perhaps a graph or a flowchart would make the process clear?

This section has been edited to make it clearer.

The manuscript doesn't distinguish between original tweets and retweets, but I'd argue that there is a difference between them. A retweet, for instance, doesn't necessary mean that the tweeter agrees with the original tweet. What was the proportion of retweets? It would be interesting to see if the results would change significantly if only original tweets were coded, but at least I think the authors should raise this question in the discussion, perhaps as a possible limitation of the study.

There is a difference between original tweets and retweets. Using Discovertext we were unable to tell the proportions of retweets/tweets. We have included this issue as a specific study limitation in the Discussion section:

Using the Discovertext software, we were unable to distinguish between original tweets and retweets. It is likely that a significant proportion of the tweets were retweets, but we are unable to gauge what proportion, and this remains a limitation of our study. While retweets are perceived by many as an expression of agreement with the original tweet, this is not always the case. On occasion, retweets were accompanied by a comment from the user. In these circumstances the sentiment of the extra comment was analysed primarily, rather than the sentiment of the retweet.

While I think that the results of this study do answer the research questions, it would benefit the reader if the results were better linked to the respective research questions, i.e. the authors could and probably should return to the research questions in the end.

Many thanks for your feedback, We have attempted to make this more explicit in the Discussion.

Reviewer: 2

Reviewer Name: Anurag Sharma

Institution and Country: University of New South Wales, Sydney, Australia

Please state any competing interests or state 'None declared': N/A

Please leave your comments for the authors below

Sin taxes are gaining popularity among policy makers as a tool for behavioural change to achieve better health outcomes in the population. However, the response to such policies can be varied due to various stakeholders involved. In particular contrasting claims are made regarding general public support for such taxes. Thus there is a need for more evidence to gauge public perception of such policies. This paper helps fill this gap by using data from social media (twitter). Data from tweets related to the MUP policy were collected over two weeks after the policy was implemented in Scotland. Despite some obvious issues related to the representativeness of such data (adequately discussed in the limitations section) 35% of tweets were found to be positive compared to 28% negative tweets. My comments:

1. Pairwise group mean t-tests could be performed to see if the difference between positive, negative and neutral tweets is statistically significant. Similarly, groupings can be done for nationality and CIs reported in the table.

We have performed statistical tests as proposed, though rather than using a t-test we have used a chi-squared test as the data are categorical:

A chi-square test found a significant difference (p -value <0.001) between observed and expected values for positive and negative tweets, but failed to find a significant difference for neutral tweets (p -value 0.079).

2. The fact that only 28% comments could be perceived as negative is the main takeaway from the results and should be emphasised in the abstract and discussion section.

We have now emphasised this both in the abstract and also in the Discussion section.

3. Page 5 line 8-12 : Discussion not clear especially: 4 positive, 8 negative

Thanks, this has been edited and hopefully it is clearer now.

4. More discussion/analysis needs to be done if negative tweets were backed by industry lobby groups explicitly/implicitly (through proxies) and if not what could be the possible reasons behind it.

This would be a very interesting avenue to explore further, however, due to the limitations of our software we are unable to do more than speculate about the role of industry in supporting astroturf movements on social media.

5. The inference that industry groups were relatively inactive post implementation of policy could not be made just by twitter data analysis as they might be pursuing other avenues to change public perception.

We have added this point to the Discussion.

6. It would have been interesting to see if there was surge in suggestions on the social media to get around MUP and somehow consume cheaper substitutes through border crossing etc. That implies positive reception of policy may not necessarily led to behaviour change. This issue is critical for the relevance of the findings: even if it is established this policy was well supported, will this lead to change in behaviour and reduced alcohol consumption. The last section should include discussion around this theme.

This is an interesting point, especially considering our finding that 14.2% of negative tweets were related to illicitly obtaining alcohol. All discussion on cross border trading and illicit sourcing of alcohol was, however, categorised as negative sentiment, and so this negativity would have been adequately represented. There is, however, the possibility that those who are initially positive may still seek to dodge MUP as you suggest. We have added this point to the Discussion:

In addition, while we demonstrated a high proportion of positive posts, this may not necessarily translate into behaviour change, or speak directly to the possible success of the policy. Nonetheless, it is possible to appreciate the divided nature of public opinion on the introduction of MUP, the nature of the sentiment around it, and key actors involved, and it will be possible to later study how this picture changes when the policy becomes more established.

VERSION 2 – REVIEW

REVIEWER	Kim Holmberg University of Turku
REVIEW RETURNED	10-Apr-2019

GENERAL COMMENTS	The authors have adequately responded to all of the comments and the manuscript can in my opinion now be moved to publication.
--

REVIEWER	Anurag Sharma UNSW, Sydney
REVIEW RETURNED	09-Apr-2019

GENERAL COMMENTS	My suggestions have been adequately addressed in the revised version and I recommend publication.
---